# Effect of *N*-acetyl-l-cysteine on Cell Phenotype and Autophagy in *Pichia pastoris* Expressing Human Serum Albumin and Porcine Follicle-Stimulating Hormone Fusion Protein

**DOI:** 10.3390/molecules28073041

**Published:** 2023-03-29

**Authors:** Yingqing Xu, Zijian Geng, Chengxi Yang, Hongwei Zhou, Yixing Wang, Buayisham Kuerban, Gang Luo

**Affiliations:** School of Biotechnology, Jiangsu University of Science and Technology, Zhenjiang 212100, China

**Keywords:** *Pichia pastoris*, autophagy, *N*-acetyl-l-cysteine, cell wall

## Abstract

*Pichia pastoris* is widely used for the production of recombinant proteins, but the low secretion efficiency hinders its wide application in biopharmaceuticals. Our previous study had shown that *N*-acetyl-l-cysteine (NAC) promotes human serum albumin and porcine follicle-stimulating hormone fusion protein (HSA-pFSHβ) secretion by increasing intracellular GSH levels, but the downstream impact mechanism is not clear. In this study, we investigated the roles of autophagy as well as cell phenotype in NAC promoting HSA-pFSHβ secretion. Our results showed that NAC slowed down the cell growth rate, and its effects were unaffected by Congo Red and Calcofluor White. Moreover, NAC affected cell wall composition by increasing chitin content and decreasing β-1,3-glucan content. In addition, the expressions of vesicular pathway and autophagy-related genes were significantly decreased after NAC treatment. Further studies revealed that autophagy, especially the cytoplasm-to-vacuole targeting (Cvt) pathway, mitophagy and pexophagy, was significantly increased with time, and NAC has a promoting effect on autophagy, especially at 48 h and 72 h of NAC treatment. However, the disruption of mitophagy receptor Atg32, but not pexophagy receptor Atg30, inhibited HSA-pFSHβ production, and neither of them inhibited the NAC-promoted effect of HSA-pFSHβ. In conclusion, vesicular transport, autophagy and cell wall are all involved in the NAC-promoted HSA-pFSHβ secretion and that disruption of the autophagy receptor alone does not inhibit the effect of NAC.

## 1. Introduction

The methylotrophic yeast *Pichia pastoris* is widely used for heterologous production of recombinant proteins such as enzymes, growth factors, and antibodies for industrial and pharmaceutical use [1,2,3]. As a host, *P. pastoris* has the advantages of both prokaryotic and eukaryotic organisms, including its capacity to reach high cell densities in defined and complex media [4], posttranslational modifications [5,6,7], and availability of the strong and tightly regulable alcohol oxidase 1 promoter [8]. Moreover, *P. pastoris* secretes a small amount of endogenous proteins, so that recombinant proteins comprise the majority of the proteins in the culture medium, a characteristic that facilitates downstream processing [9]. Although *P. pastoris* is an efficient production platform, the yield of recombinant protein in culture medium can be further increased by enhancing the secretion capacity [10].

In order for secretion, recombinant protein enters the lumen of the endoplasmic reticulum (ER) mediated by the N-terminal α-factor secretion signal. With the help of molecular chaperones, such as Kar2 and other folding enzymes, proteins that enter the ER are folded properly [11,12]. If some recombinant proteins fail to fold properly, they will be degraded by the ER-associated degradation (ERAD) system [13,14]. Cellular engineering including overexpression of folding helpers such as protein disulfide isomerase (Pdi1), and gene optimization such as substitution of N-terminal signal peptides are efficient approaches to promote recombinant protein secretion [15,16]. Recently, Marsalek L. et al. have found that some recombinant proteins are undesired transport to the vacuole where proteins are degraded and recycled [17,18]. Blocking protein mistargeting to vacuole by disrupting the vacuolar protein-sorting component of HOPS (homo-typic fusion and protein sorting) or CORVET (class C core vacuole/endosome tethering) complex can efficiently improve recombinant secretion [17]. Thus, the secretion capacity of *P. pastoris* can be further increased by manipulating pathways [19].

In addition to molecular strategies, culture condition optimization offers a practical alternative. Our previous study has reported that supplementation with thiol-reducing reagents, such as *N*-acetyl-l-cysteine (NAC) and glutathione (GSH), improved the secretion of recombinant human serum albumin and porcine follicle-stimulating hormone fusion protein (HSA-pFSHβ) by increasing intracellular GSH level [20]. In *Saccharomyces cerevisiae*, the antioxidant NAC prevented the autophagy-dependent delivery of mitochondria to the vacuoles (mitophagy) [21]. The beneficial effect of NAC on HSA-pFSHβ production is similar to its effect on mitophagy, i.e., the effect of NAC on HSA-pFSHβ production or mitophagy is involved in its fueling effect of the glutathione pool rather than its scavenging capacity [21]. Based on these findings, we were intrigued whether the mechanism of NAC beneficial effect of HSA-pFSHβ was related to autophagy.

In the present study, we explored the effect of NAC on cell phenotype and autophagy in high-level expressing HSA-pFSHβ strain. Our results have shown that NAC supplementation reduced cell growth rate, which may be associated with increased cell wall thickness and intracellular ROS level. Moreover, the expression level of vesicular transport and autophagy related genes were significantly decreased after NAC treatment. Furthermore, autophagy including Cvt pathway, pexophagy, and mitophagy was significantly increased with inducing time and NAC has a promoting effect on autophagy. In addition, the effect of NAC on HSA-pFSHβ secretion is independent of autophagy. Our study provides a theoretical basis for uncovering the role of autophagy in promoting secretion of recombinant proteins.

## 2. Results

### 2.1. Effect of NAC on Cell Growth and Phenotype

Previous studies have shown that NAC could improve the secretion of HSA-pFSHβ by improving intracellular GSH pool [20]. To further explore the downstream pathways by which NAC enhances HSA-pFSHβ secretion, we analyzed the effect of NAC on the cell wall of F strain that high-level produced HSA-pFSHβ. As shown in Figure 1a, compared with the control group, the number of yeast clones showed a decreasing trend on Yeast extract Peptone Dextrose medium (YPD) with or without 30 μg/mL Congo Red (CR) or Calcofluor White (CFW) after 72 h of NAC treatment. As reagents interfering with the assembly of the cell wall, CR or CFW did not affect the effect of NAC on cell phenotype (Figure 1a), suggesting that NAC does not disrupt the assembly of the cell wall. To further explore the effect of NAC on the cell wall, we knocked out the Yapsin 1 (*YPS1*) gene in the F strain, which is important for the survival of thermal and cell wall stress in fungi [22]. By calculating the number of clones at dilutions of 10^−4^ and 10^−5^ in Figure 1a, we found that NAC significantly affected the number of clones of the Δ*yps1* strain that knocked out *YPS1* gene in the F strain on YPD or YPD+CR plates, but not on F strain (Figure 1b,c). In short, NAC supplementation inhibited the cell growth rate of F strain, whereas CR or CFW reagents did not interfere with the effects of NAC.

### 2.2. NAC Affects the Composition of the Cell Wall

As a glycosylphosphatidylinositol-linked aspartyl protease, yps1 plays an important role in cell wall integrity [23]. To further test the effect of NAC on the cell wall, we examined β-1,3-glucan and chitin content, which is the essential composition of the cell wall. The chitin contents were significantly increased [24] and the β-1,3-glucan contents were significantly reduced in F strain and Δ*yps1* strains supplemented with NAC, compared to the control (Figure 2a,b). The transmission electron microscopy experiment further verified our experimental results. As shown in Figure 2c, compared to control, the space between the cell membrane and the cell wall became smaller and the cell wall was thickened with NAC treatment. These results provide us with the information that NAC strengthens the thickness of the cell wall.

Substantial changes in cell wall composition and thickness also occur in response to environmental physical stress, such as osmotic and heat stresses, to avoid cell membrane rupture and lysis [25]. To further investigate the role of stress in the alteration of cell wall composition by NAC, we examined intracellular reactive oxygen species (ROS) levels after 72 h of NAC treatment. As shown in Figure 2d, the fluorescence intensity of 2′,7′-dichlorodihydrofluorescein diacetate (DCFH-DA), a probe to detect ROS, was significantly enhanced by NAC supplementation.

### 2.3. Transcript Expression Analysis of Genes Involved in Vesicular Trafficking

Our previous work has shown that increasing ER processing efficiency did not enhance HSA-pFSHβ secretion [26]. Moreover, engineering the early secretory pathway is an alternative method to increase the recombinant protein secretion [19]. Based on the above results, we speculated that the target of NAC on *P. pastoris* may be located in the vesicle trafficking. Next, the expression levels of genes encoding proteins related to the vesicular transport pathway were tested by qRT-PCR.

As shown in Figure 3, compared with control group, the expression levels of all genes except *GET 3*, *SEC 4*, *TRS 23*, *SRP 9*, and *SEC 2*, showed a decreasing trend, and the expression levels of 13 of them were significantly decreased (all primers used in this study are shown in Appendix A). Among them, a large proportion of genes (*PEX 14*, *ATG 9*, *ATG 13*, *ATG 2*, *PEX 3*, *ATG30* and *ATG32*) coding products are involved in autophagy. Moreover, *PEX 14*, *PEX 3*, and *ATG30* coding products are involved in pexophagy [27], and *ATG30* and *ATG32* coding products are pexophagy and mitophagy receptors, respectively [28]. Thus, autophagy, especially mitophagy and pexophagy, is involved in the promotion of recombinant protein secretion by NAC. In addition, the encoded products of four other genes (*SAR 1*, *ARF 1*, *SNC 2*, *COG 4*) play important roles in vesicular trafficking associated with the Golgi apparatus [29,30,31].

### 2.4. Autophagy Is Involved in NAC-Promoted HSA-pFSHβ Secretion

Based on the above results, we further assayed the intracellular alcohol oxidase (AOX) and vacuolar enzyme carboxypeptidase Y (CPY) activity associated with autophagy. As shown in Figure 4a, the activity of intracellular CPY did not significantly change with NAC supplementation. Moreover, the AOX activity in F and Δ*yps1* strains treated with NAC was significantly decreased compared to the control (Figure 4b). It has been reported that the activity of AOX can be used as an indicator to monitor pexophagy [27]. The decrease in AOX activity indicated that autophagy is increased in F and Δ*yps1* strains dealt with NAC.

To further confirm the effect of NAC on autophagy, we overexpressed the plasmid pGAPZA-yEGFP3-ATG8 in F strain (named FATG8 strain, all recombinant strains used in this study are shown in Table 1). During autophagy, green fluorescence protein (GFP) fusion protein is delivered to the vacuole, where the GFP moiety is cleaved off by vacuolar hydrolases. GFP is more stable than GFP-fusion protein in vacuole. As shown in Figure 4c, the upper and lower protein bands are yEGFP3-ATG8 fusion protein and yEGFP3, respectively. Compared to 0–24 h, the upper protein band (indicated by arrow 1) was significantly decreased and the lower protein band (indicated by arrow 2) was significantly increased after 48 and 72 h of induction. (Figure 4c). These results indicated that autophagy was enhanced at later stages of HSA-pFSHβ protein production. Moreover, we compared the ratio of yEGFP3 and yEGFP3-ATG8 based on protein band density in arbitrary units as quantified by densitometry analysis in Figure 4c. Compared with the FATG8 strain without NAC supplementation, the ratio of yEGFP3/yEGFP3-ATG8 was significantly increased after 48 and 72 h of NAC treatment, indicating that NAC enhanced autophagy of FATG8 strain (Figure 4d). In addition, compared to that in BMGY medium, yEGFP3/yEGFP3-ATG8 ratio was decreased at 0 h in BMMY medium, indicated that the autophagy was reduced after transferring from BMGY to BMMY (Figure 4e).

### 2.5. The Types of Autophagy Affected by NAC

To further confirm which types of autophagy are involved in the activity of NAC on recombinant protein yield, we monitored the nonselective and selective autophagy by Western blot. As a marker for nonselective autophagy, aminopeptidase I precursor (prApe1) can be detected by fusing yEGFP3 protein using GFP antibody. As shown in Figure 5a, the Ape1-yEGFP3 protein band was only observed upon overexposure. Nevertheless, only faint protein bands were observed after 24 h of HSA-pFSHβ-induced expression. The results of grayscale analysis showed that yEGFP3/Ape1-yEGFP3 ratio gradually increased with induction time until it started to decrease at 96 h, which corresponded to the yield of recombinant protein HSA-pFSHβ in the culture medium of FApe1 strain (Figure 5b and Appendix A). Compared to the FApe1 strain without NAC supplementation, the yEGFP3/Ape1-yEGFP3 ratio was significantly increased after 6, 24, 48 and 72 h of NAC treatment and decreased after 96 h of treatment. Moreover, a similar trend of increased yEGFP3/Ape1-yEGFP3 ratio was observed after 12 h of NAC treatment (1.75 ± 0.11 of NAC group versus 1.61 ± 0.08 of control). The yield of HSA-pFSHβ in culture medium of FApe1 strain treated with NAC was significantly increased at 24, 48 and 72 h, compared to that without NAC supplementation (Appendix A). These results suggested that autophagy is involved in the process of NAC-promoted HSA-pFSHβ secretion. 

In addition, we observed the localization of Ape1 after 72 h of NAC treatment of FApe1 strain. As shown in Figure 5c, Ape1-yEGFP3 was distributed in the cytoplasm outside the vacuole in a punctate form, and yEGFP3 was distributed in the vacuole in pieces form. The punctate Ape1-yEGFP3 fusion protein entered the vacuole to degrade into yEGFP3, and the total fluorescence intensity became brighter after NAC treatment (Figure 5c). Moreover, the total fluorescence intensity was significantly increased in FApe1 strain treated with NAC (Figure 5d). These results indicated that NAC increased the nonselective autophagy during HSA-pFSHβ production.

Next, we monitored the selective autophagy, including mitophagy and pexophagy. As shown in Figure 6a, three protein bands could be detected by GFP antibody. The upper band was Tom20-yEGFP3 fusion protein, the lower protein band was yEGFP3 protein, and the middle band might be the truncated Tom20-yEGFP3 fusion protein. The yEGFP3 protein band was only detected after 24 h of HSA-pFSHβ induction. The ratio of degraded fragments (including middle and lower protein band) to Tom20-yEGFP3 fusion protein band was used to quantify the western blot result. The degradation of Tom20-yEGFP3 fusion protein was significantly increased after 6, 9, 12, 48 and 72 h of NAC treatment (Figure 6b). As with that in FApe1 strain, the degradation of Tom20-yEGFP in the FTom20 strain was synchronized with the increase of HSA-pFSHβ yield in the culture medium, indicating that mitophagy is involved in NAC-promoted secretion of HSA-pFSHβ (Appendix A).

Moreover, we localized peroxisomes with BFP-SKL and observed the proportion of peroxisomes in the vacuoles after 48 h of NAC treatment. As shown in Figure 6c, the blue fluorescent signal in the vacuole is brighter than that in the cytoplasm (marked with white arrow), which is caused by the accumulation BFP. We found the presence of peroxisomes within vacuoles was significantly increased compared to the control group (37.19% in NAC-treated group versus 30.54% in the control group) by counting the cells with peroxisomes in vacuoles. These observations suggested that NAC affects the pexophagy in F strain during HSA-pFSHβ production, which was consistent with the previous result of AOX activity.

### 2.6. Role of Autophagy in the Promotion of HSA-pFSHβ Secretion by NAC

To investigate whether NAC promotes HSA-pFSHβ secretion by increasing autophagy, we disrupted the mitophagy and pexophagy receptors, Atg32 and Atg30, respectively. As shown in Figure 7a, an adenine was deleted at 316-base-pair of the *ATG30* gene and 539-base-pair of *ATG32* gene, resulting in frameshift mutation of *ATG30* and *ATG32* genes in F strains, named F∆Atg30 and F∆Atg32, respectively. To detect the effect of *ATG30* and *ATG32* gene disruption on recombinant production, we detected the HSA-pFSHβ protein band of F∆Atg30 and F∆Atg32 in the cell and culture medium supplemented with or without NAC. As shown in Figure 7b, the intact HSA-pFSHβ and truncated HSA protein bands could be detected in culture medium of F∆Atg30 strain. Moreover, compared with control, recombinant HSA-pFSHβ was obviously increased by supplementing NAC, which is consistent with our previous study, indicating that *ATG30* gene disruption has little effect on the role of NAC in promoting HSA-pFSHβ. Unlike *ATG30*, *ATG32* gene disruption significantly affected the HSA-pFSHβ yield in culture medium, because HSA-pFSHβ and truncated HSA could not be detected on Coomassie brilliant blue-stained 12% SDS-PAGE gel. To further explore the effect of *ATG32* gene disruption on HSA-pFSHβ production, we tested HSA-pFSHβ protein band in cell and culture medium supplemented with NAC by Western blot. The yield of HSA-pFSHβ in culture medium was significantly increased, while HSA-pFSHβ production in cells was significantly decreased, indicating that *ATG32* gene disruption does not significantly affect the effect of NAC in promoting HSA-pFSHβ secretion (Figure 7c). In short, our results indicated that mitophagy and pexophagy receptor are involved in NAC-promoted HSA-pFSHβ secretion in an independent manner.

## 3. Discussion

The methylotrophic yeast *P. pastoris* is a protein expression host applied for the production of recombinant proteins. However, because of inefficient secretion, many recombinant proteins, particularly therapeutic proteins, have poor yields by *P. pastoris* [32,33,34]. Improving secretion levels through medium optimization or engineering strains may be a reasonable and promising strategy to increase the production of heterologous proteins by *P. pastoris* [35,36,37]. Our previous study has reported that NAC can promote HSA-pFSHβ secretion by increasing intracellular GSH pools [20]. However, the downstream regulatory pathway by which NAC promotes HSA-pFSHβ secretion via GSH remains elusive.

Based on the previous finding that enhancing ER processing efficiency did not enhance HSA-pFSHβ secretion [26], we examined the transcript levels of vesicle-associated genes among Golgi, endosome, and ER, to determine whether downstream vesicle transport affects recombinant HSA-pFSHβ secretion. As small GTPases, Sar1 and Arf1 protein bidirectionally manage vesicular trafficking in the early secretory pathway between the ER and Golgi. The transport of COPII-coated vesicles (from ER to Golgi) and COPI-coated vesicles (from Golgi to ER) depends on Sar1 and Arf1, respectively [29]. Another Rab GTPase, Vps 21 mediates CORVET from trans-Golgi Network (TGN) to early endosomes by binding to the CORVET complex. Moreover, Snc2 functions for the endosome-derived vesicular fusion at the TGN [30], and Cog 4 is a member of COG complex located on Golgi, which responses to transport vesicle from endosome to cis Golgi [31]. The reduced expression level of *SAR1*, *ARF1*, *SNC2, VPS21,* and *COG4* genes indicated that NAC affects vesicular transport among Golgi, endosome, and ER.

In addition, recombinant protein may mistarget to vacuole for degradation through vesicular transport [18]. Disruption of proteins associated with mistargeted vesicles is one of the effective ways to improve the protein secretion [18]. Our results implied that NAC reduces the mistargeting of nascent HSA-pFSHβ proteins to vesicles, leading to down-regulation of the expression of vesicular transport related genes, including *YDJ 1* whose coding product Ydj 1 plays an important role in the translocation process, such as GPI-anchored proteins [38].

Autophagy denotes a group of highly conserved catabolic pathways that transport intracellular components including recombinant protein to the vacuole/lysosome for their degradation [39]. In addition to genes encoding vesicle transport-related proteins, we also found significant downregulation of autophagy proteins coding genes. These proteins include receptors of pexophagy and mitophagy (two selective autophagy), peroxisomal membrane proteins Pex 3 and Pex 14, and other essential autophagy proteins (Atg 9, Atg2, Atg13, and Vps21) [27,28,40,41,42].

Autophagy can be divided into nonselective and selective autophagy, the latter including mitophagy, pexophagy, aggrephagy, ER autophagy, and so on. Environmental factors such as starvation, oxidative stress, or chemical agents can induce autophagy in yeast and mammalian cells [43,44,45]. In *P. pastoris*, pexophagy was induced by transferring cultures on non-fermentative carbon sources (such as methanol or oleate) to glucose. In this study, we proliferated the yeast with BMGY medium (glycerol as carbon source) and then transferred it to BMMY medium (methanol as carbon source) for induction. We used ATG8, that affects autophagosome size, to the observed autophagy [46]. During the transformation of cells from glucose medium to methanol medium, autophagy was decreased, and then gradually increased as the strains were induced in BMMY for a longer period. These results were consistent with previous results reported by Yamashita et al. [47]. In addition, the Cvt pathway, mitophagy, and pexophagy increased in the late stages of induction of HSA-pFSHβ protein by detecting autophagy markers and localized peroxisomes, which corresponds to an increase of HSA-pFSHβ. Thus, autophagy is involved in the secretion of HSA-pFSHβ. NAC further increases the production of recombinant HSA-pFSHβ in the culture medium by increasing autophagy.

Peroxisomes are single-membrane-bound organelles whose functions are to scavenge reactive oxygen species (ROS) and to catalyze fatty acid β-oxidation [48]. The AOX activity and peroxisomes localized assay indicated that pexophagy was increased after 72 h of NAC treatment, leading to an increase in ROS levels. It has been reported that NAC could alleviate mitophagy without increasing ROS level in *Saccharomyces cerevisiae* [21]. Our experiment has the similar results that NAC increases ROS level by degrading peroxisomes through pexophagy. Moreover, the beneficial effect of NAC on HSA-pFSHβ production was related to increasing the Cvt pathway, mitophagy and pexophagy rather than alleviated mitophagy in *P. pastoris*. The different effects of NAC on yeast may be related to the way autophagy is induced. In *Saccharomyces cerevisiae*, nitrogen starvation was used to induce autophagy. In this study, autophagy was induced in the inducing stage of the HSA-pFSHβ protein. In both conditions, the function of NAC was maintained in autophagic homeostasis by increasing GSH levels.

Mitophagy and pexophagy receptors Atg 32 and Atg 30 knockdown experiments showed that autophagy is a non-major factor in the promotion of HSA-pFSHβ secretion by NAC. In addition to autophagy, we also found that NAC reduced cell growth rate without affecting cell apoptosis (Appendix A) and affected cell wall component. It has been reported that the thickness of cell wall was increased by reducing β-1,3-glucan level and increasing chitin content through disrupting fks1 [49]. NAC has a similar effect on the cell wall. A high chitin content can contribute to the strength of the cell. Thus, increased thickness of cell wall would make *P. pastoris* cell more resistant to the pressure from the environment. In addition, the signaling of cell wall integrity is coordinated with autophagy [50].

## 4. Conclusions

In conclusion, our study confirmed that autophagy is involved in the secretion of recombinant HSA-pFSHβ in an independent manner. NAC promotes HSA-pFSHβ secretion by regulating the vesicular transport pathway and autophagic pathway, increasing cell wall thickness, and inhibiting cell growth. In contrast, inhibition of pexophagy did not affect the effect of NAC on HSA-pFSHβ secretion, and inhibition of mitophagy could affect the secretion of HSA-pFSHβ, but not the role of NAC. Thus, autophagy is involved in NAC-promoted HSA-pFSHβ secretion, but NAC does not depend on autophagy in this action. Our results provide a theoretical basis for studying autophagy occurring during recombinant protein secretion in *P. pastoris*.

## 5. Materials and Methods

### 5.1. Strains, Plasmids, and Reagents

*P. pastoris* strain GS115-pPIC9K-HSA-pFSHβ (Termed F strain, HSA-pFSHβ gene ID: MH249035) was constructed in our laboratory [34], *Escherichia coli* strain Top10 (CWbio, Beijing, China), and vectors pGAPZA, and pPICZA (Invitrogen, Shanghai, China) were used for cloning and heterologous expression. N-acetyl-L-cysteine (NAC, Cat#A7250) was provided by Sigma-Aldrich (St. Louis, MA, USA). Restriction enzymes *Eco* RI*, Spe* I*,* and *Sal* I were obtained from TAKARA Biotechnology Co., Ltd. (Dalian, China), *Pme* I, *Ban* II and *Avr* II was purchased from New England Biolabs (Beijing, China). Zeocin was purchased from Thermo Fisher Scientific (Waltham, MA, USA).

### 5.2. Construction of Expression Plasmids of pGAPZA-yEGFP3-ATG8, pGAPZA-BFP-SKL, pACTZ-Ape1 and pTom20-yEGFP3

The coding gene of yeast enhancing green fluorescent protein mutant 3 (yEGFP3) [51] (GenBank: U73901.1) and monomeric blue fluorescent protein (GenBank: AZQ25074.1) with SKL tripeptide at C-terminal (BFP-SKL) were optimized and delivered to GenScript Biotech Corp. (Nanjing, China) for de novo synthesis. *ATG 8*, *APE 1* and *TOM 20* genes were amplified from GS115 genome by using corresponding primers in Appendix A. The digestion product of yEGFP3, ATG8 and BFP-SKL ligated into linearized plasmid pGAPZA cut with *Eco* RI/*Sal* I to construct pGAPZA-yEGFP3-ATG8 and pGAPZA-BFP-SKL plasmid. pACTZ-Ape1 and pTom20-yEGFP3 were constructed by homologous recombination using ClonExpress Ultra One Step Cloning Kit (Vazyme Biotech, Nanjing, China).

### 5.3. Transformation and Screening Recombinant Strains

The plasmids pGAPZA-yEGFP3-ATG8, pTom20-yEGFP3, pACTZ-Ape1, and pGAPZA-BFP-SKL linearized using *Avr* II, *Spe* I, *Ban* II and *Avr* II, respectively were transformed into *P. pastoris* strain GS115-pPIC9K-HSA-pFSHβ by electroporation. Positive transformants were selected directly on YPD+zeocin (Invitrogen, Shanghai, China) plates (1% yeast extract, 2% peptone, 2% dextrose, 2% agar, and 300 μg/mL zeocin) for 3–5 days at 30 °C and identified by direct PCR [52].

### 5.4. Construction of ΔAtg30 and ΔAtg32 Disrupted Strains

Δ*Atg30* and Δ*Atg32* disrupted strains were constructed by using CRISPR/Cas 9-based homology-directed genome editing as described by Gassler [52]. In brief, a single guide RNA of *ATG30* and *ATG32* gene (underlined parts of 2-ATG30-sgRNA-fw1 and 2-ATG32-sgRNA-fw1 in Appendix A) was designed based on a protospacer adjacent motif (PAM) sequence identified in 50–200 bp upstream of the *ATG30* and *ATG32* gene CDS on website (http://chopchop.cbu.uib.no/). The ribozyme-sgRNA-fusion gene was generated by overlap extension PCR of six primers (Structural primers A-sgRNA-struc-rev, B-sgRNA-struc-rev, C-sgRNA-struc-rev, D-sgRNA-struc-fw with primers 1-ATG30-sgRNA-fw1 + 2-ATG30-sgRNA-fw1 or 1-ATG32-sgRNA-fw1 + 2-ATG32-sgRNA-fw1) as shown in Appendix A. For CRISPR/Cas9-ΔAtg30 and CRISPR/Cas9-ΔAtg32 plasmids (BB3cH_pGAP_23 *_pPFK300_Cas9 plasmid containing the sgRNA of *ATG30* and *ATG32* gene, respectively), the Golden Gate assembly method was performed by using restriction enzyme *Bpi* I (Themo Fisher Scientific, Waltham, MA, USA). The circular CRISPR/Cas9-ΔAtg30 and CRISPR/Cas9-ΔAtg32 plasmids DNA were transformed in F strain by electro-transformation following the Pichia expression system manual (Thermo Fisher Scientific) with minor modification. Briefly, we extended the ice bath time of circular CRISPR/Cas9 plasmids and electro-competent cells mixture to 20 min, and the incubation time to 3 h after transformation. Δ*Atg30* and Δ*Atg32* disrupted strains were checked by direct PCR using primers gap-ATG30-F + ATG30-R and ATG32-EcoRI-F + ATG32-yEGFP3-R, respectively, and sequencing. After confirmation of the deletion of ATG30 and ATG32 genes, true KO strains were passaged at least three times on YPD to lose the CRISPR/Cas9-ΔAtg30 and CRISPR/Cas9-ΔAtg32 plasmids. The positive strains were named FΔAtg30 and FΔAtg32.

### 5.5. Proteins Expression and Analysis

Recombinant HSA-pFSHβ protein was induced according to the previously reported method [22]. The recombinant HSA-pFSHβ protein in 5 μL culture medium or 20 μg of total protein were analyzed using 12% SDS-PAGE and western blot. For Western blot, a mouse anti-human FSHβ monoclonal antibody and GFP antibody (Santa Cruz Biotechnology, Santa Cruz, CA, USA, 1:500) were used as the primary antibody to detect pFSH and yEGFP3, respectively. The secondary antibody (Zhongshan jinqiao Biotech, Beijing, China, diluted 1:5000–1:10,000) was horseradish peroxidase (HRP) conjugated goat anti-mouse IgG. The immunoreactive proteins on the blots were visualized with ECL and imaged on an ImageQuant LAS 4000 instrument. The band intensities were quantified using Image J software. Western analysis data are representative of triplicate experiments. For autophagy determination, the recombinant strains (including FATG8, FApe1, and FTom20) were activated in 3 mL BMGY medium for 48 h. Cells were then transferred to 1 mL BMMY medium for induction for 96 h, at which point NAC was added and noted as 0 h. Methanol was added every 24 h.

### 5.6. Phenotypic Analysis

5 × 10^7^ cells were collected at 72 h of induction and diluted in a series of 10-fold magnitude (from 10^−1^ to 10^−5^ relative to the initial culture) in water. Aliquots (2 μL) of 10-fold serial dilutions were spotted onto the YPD plates with or without 30 μg/mL Calcofluor White (CFW) or Congo Red (CR). Photographs were taken after 3 days of incubation in 30 °C incubator. There were three repeats for each strain.

### 5.7. Quantitative Chitin and β-1,3-glucan Measurement

The cell wall chitin and β-1,3-glucan contents were measured according to the previously reported method [22]. The fluorescence density (FLU) of the cells was determined using SpectraMax i3 fluorescence microplate reader. The excitation and emission wavelengths for measuring chitin were 325 nm and 435 nm, respectively, and the excitation and emission wavelengths for measuring β-1,3-glucan were 386 nm and 460 nm, respectively.

### 5.8. Quantitative Real-Time PCR

Cells of F strain dealt with or without 5 mM NAC for 72h was collected, and total RNA was extracted using the TRIzol reagent (Invitrogen, Shanghai, Beijing) and 1 μg of total RNA was reverse transcribed using HiScript™ Q RT SuperMix for qPCR (+gDNA wiper) (Vazyme Biotech Co., Ltd., Nanjing, China). Quantitative real-time PCR was performed using a CFX96 Real Time system (Bio-Rad, Beijing, China) in a reaction mixture containing (in a total volume of 10 μL) 2 × Taq Pro Universal SYBR qPCR Master Mix (Vazyme, Nanjing, China), forward and reverse primers, cDNA, and ddH_2_O. PCR was performed at 95 °C for 30 s, followed by 39 cycles of 95 °C for 10 s and 60 °C for 30 s, and a melting curve was constructed at the end of the amplification. The mRNA levels of trafficking-related genes were determined using qPCR analysis with the primers shown in Appendix A. Bio-Rad CFX Maestro software was used to calculate normalized gene expression values by the ΔCT method, with β-actin as the reference gene. For accurate quantification of RT-PCR products, at least three technical replicates of three biological replicates were obtained. Data were statistically analyzed by nonparametric tests, and a *p* < 0.05 was considered significant.

### 5.9. Determination of Alcohol Oxidase (AOX)

The AOX activity in cells of F and ∆*yps1* strains were measured as Schroder [53] with slight modification. Briefly, the cells collected after 72 h induction were lysed by Tam [54]. The enzyme extracts were obtained by 2860× *g* for 10 min at 4 °C, and then incubated with the reactive buffer (10 U/mL horseradish peroxidase, 1 mmol/L 4-aminoantipyrine, 4.3 mol/L phenol, and 200 mmol/L methanol in 10 mmol/L PBS (pH7.5)), for 10 min at 37 °C. The OD_500 nm_ of reactive product was evaluated using Epoch2 microplate reader (Bio-Tek Instruments, Winooski, VT, USA). One unit of AOX activity was defined as the amount of enzyme required to produce 1 mmol hydrogen peroxide per minute.

### 5.10. Intracellular ROS Measurement

The intracellular ROS levels were measured by using a Reactive Oxygen Species Assay Kit (Beyotime Biotechnology, Shanghai, China). Briefly, the cells dealt with or without 5 mM NAC were harvested by centrifugation at 3000× *g* for 3 min after inducing for 72 h and washed twice in distilled PBS. OD_600 nm_ was measured and 5 × 10^7^ cells were incubated with DCFH-DA in BMMY for 20 min at 30 °C and then measured at 488 nm excitation wavelength and 525 nm emission wavelength by a fluorescence spectrophotometer (BioTek, Winooski, VT, USA).

## Figures and Tables

**Figure 1 molecules-28-03041-f001:**
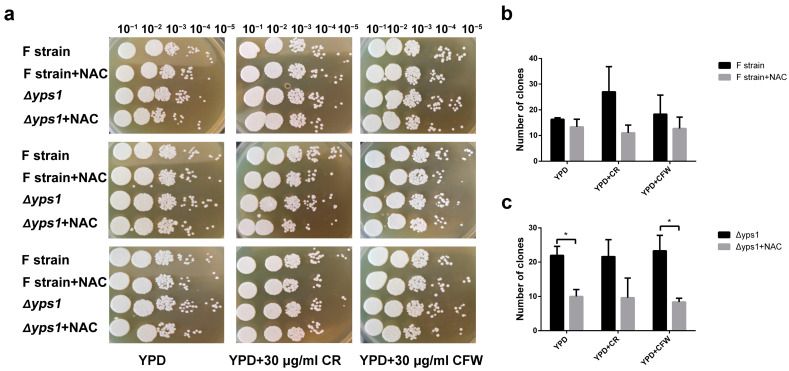
Effect of NAC on the cell growth and phenotype. (**a**) Effect of NAC on the growth of F and Δ*yps1* strains and the sensitivity of cells to 30 μg/mL CR and CFW after NAC treatment. F and Δ*yps1* strains were collected after 72 h of NAC treatment and normalized to an OD_600 nm_ of 1.0. The cells were diluted five times in a 10-fold series, and 2 μL of each dilution were spotted on YPD plates containing 30 μg/mL of CR or CFW. The same row represents different treatments, and the same column represents three technical repetitions. (**b**,**c**) Statistical analysis of the number of clones at 10^−4^ and 10^−5^ on (**a**). *p* values were calculated using Student’s two-tailed *t*-test with *p* < 0.05 considered statistically significant (Marked with *). Error bars represent means ± standard deviation (SD) (n = 3).

**Figure 2 molecules-28-03041-f002:**
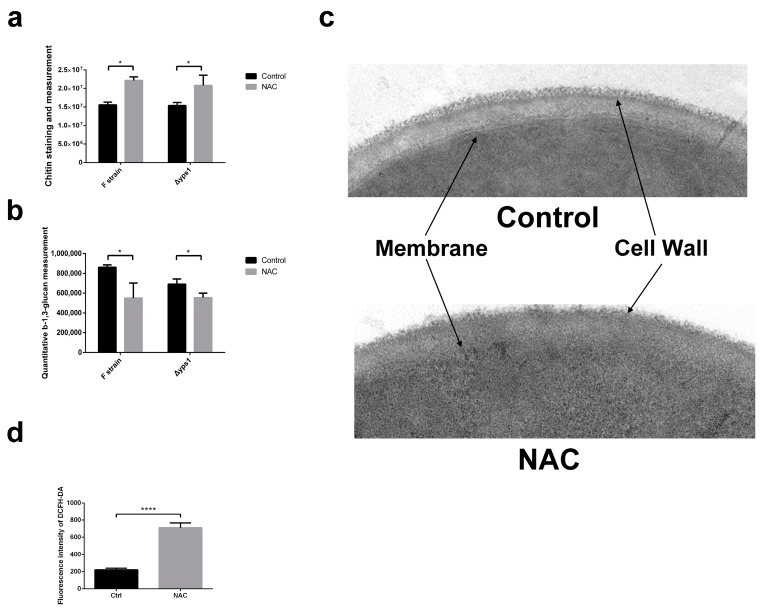
Effect of NAC on the cell wall composition. (**a**) Chitin staining and measurement. F and Δ*yps1* strains dealt with or without NAC were collected by centrifugation at 3000× *g* for 5 min, and cells were stained with 20 μg/mL CFW. (**b**) Quantitative β-1,3-glucan measurement. The β-1,3-glucan content in the cell wall was measured with aniline blue. (**c**) Transmission electron microscopy (TEM) of the F strain treated with 5 mM NAC for 72 h. CW: cell wall, Mem: cell membrane. (**d**) Total ROS levels of 5 × 10^7^ cells were measured with 2′,7′-dichlorofluorescin diacetate (DCFDA) reagent after 72 h of NAC treatment. Cells were collected and washed twice with PBS, and then incubated with BMMY containing 10 μM DCFDA at 30 °C. There were three repeats for each group, and *p* values were calculated using Student’s two-tailed *t*-test with *p* < 0.05 considered statistically significant (Marked with *), and **** represent extremely significant difference (*p* < 0.01). Error bars represent means ± standard deviation (SD) (n = 3).

**Figure 3 molecules-28-03041-f003:**
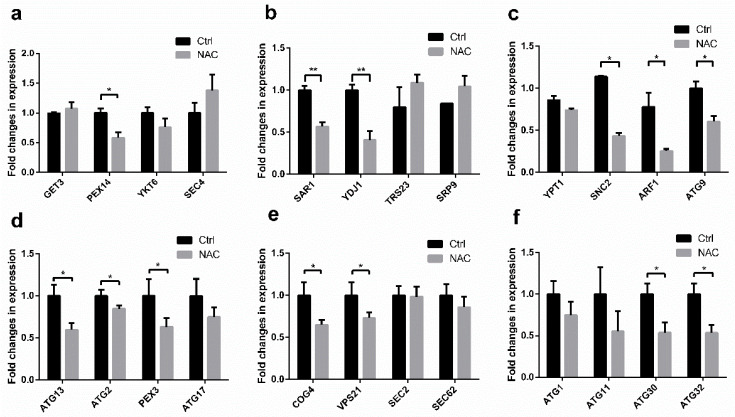
Effect of NAC on the mRNA expression levels of genes involved in vesicular trafficking and autophagy. (**a**–**c**) Relative transcription of vesicular trafficking-related genes supplemented with or without NAC. (**d**–**f**) Relative transcription of autophagy-related genes supplemented with or without NAC. All gene expression levels were relative to β-actin and normalized to non-NAC-treated control (Ctrl). All primers used in this study are shown in Appendix A. Each bar represents the mean value from three determinations including the standard deviations. * indicates statistical significance of *p* < 0.05 (Marked with *) and ** represent extremely significant difference of *p* < 0.01 by using Student’s two-tailed *t*-test.

**Figure 4 molecules-28-03041-f004:**
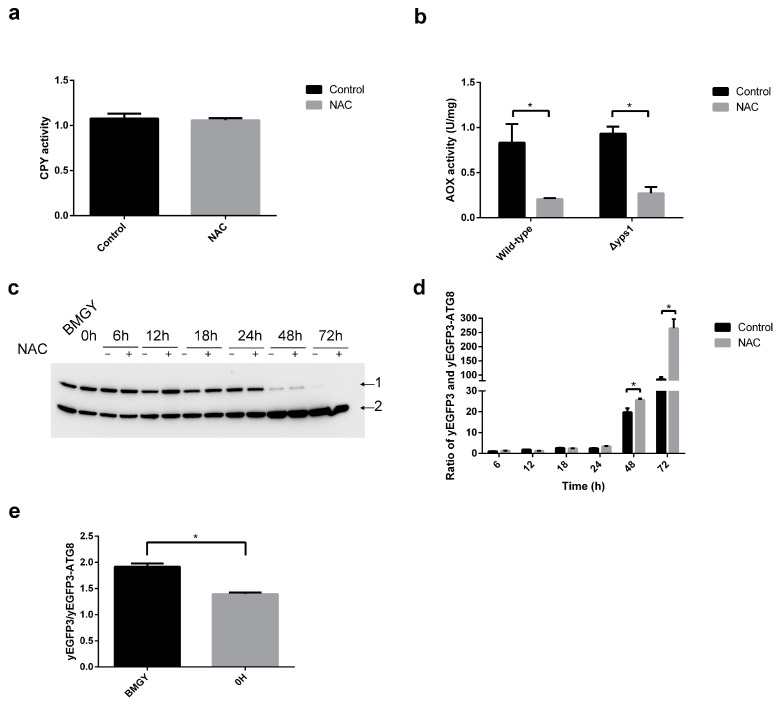
Determination of autophagy in cell treated with NAC. (**a**) Carboxypeptidase Y (CPY) activity detected in F strain treated with NAC. (**b**) Alcohol oxidase (AOX) activity measurement. The activity of AOX in F and Δ*yps1* strains supplemented with or without NAC was measured. (**c**) Western blot analysis of GFP in FATG8 strain that overexpressed yEGFP3-ATG8 in F strain. Moreover, 0 h is defined as the time of cells transfer into the BMMY. Arrow 1: yEGFP3-ATG8, arrow 2: yEGFP3. (**d**,**e**) The relative amounts of protein bands in Figure 4c were estimated using Image J analysis software. The results presented in the histogram reflect the fold-change in yEGFP3 and yEGFP3-ATG8. *p* values were calculated using Student’s two-tailed *t*-test with *p* < 0.05 considered statistically significant (Marked with *). Error bars represent means ± standard deviation (SD) (n = 3).

**Figure 5 molecules-28-03041-f005:**
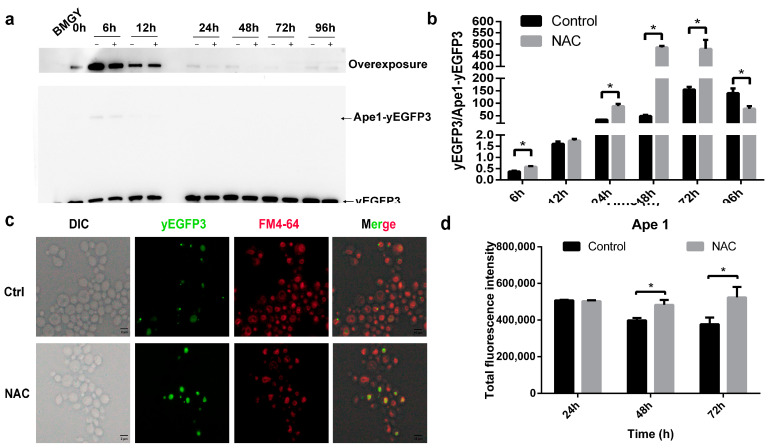
Effect of NAC on the Cvt pathway in FApe1 strain. (**a**) Western blot analysis of GFP in FApe1 strain that overexpressed Ape1-yEGFP3 in F strain. (**b**) The relative amounts of protein bands in Figure 5a were estimated using Image J analysis software. The results presented in the histogram reflect the fold-change in yEGFP3 and Ape1-yEGFP3. *p* values were calculated using Student’s two-tailed *t*-test with *p* < 0.05 considered statistically significant (Marked with *). Error bars represent means ± standard deviation (SD) (n = 3). (**c**) yEGFP3 and yEGFP3 fused proteins were observed using a fluorescence microscope in FApe1 strain after 72 h of NAC treatment. (**d**) Total fluorescence intensity of FApe1 strain treated with NAC for different times was measured by using SpectraMax i3 fluorescence microplate reader with an excitation wavelength at 485 nm, and emission wavelength at 535 nm. *p* values were calculated using Student’s two-tailed *t*-test with *p* < 0.05 considered statistically significant (Marked with *). Error bars represent means ± standard deviation (SD) (n = 3).

**Figure 6 molecules-28-03041-f006:**
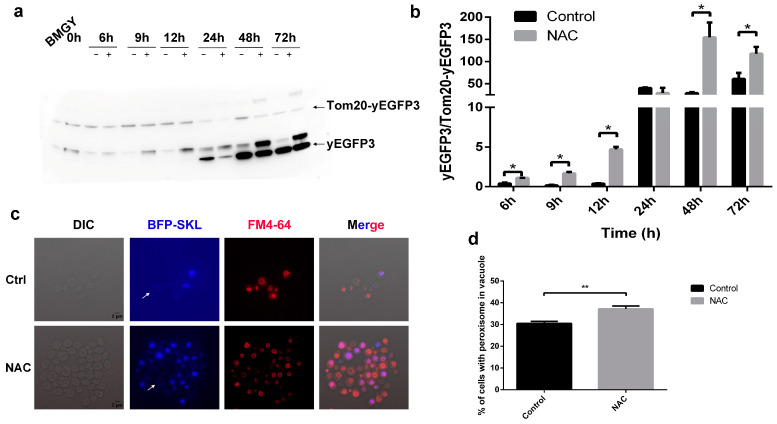
Effect of NAC on mitophagy and pexophagy. (**a**) Western blot analysis of GFP in FTom 20 strain that overexpressed Tom 20-yEGFP3 in F strain. (**b**) The relative amounts of protein bands in Figure 6a were estimated using Image J analysis software. The results presented in the histogram reflect the fold-change in yEGFP3 and Tom 20-yEGFP3. *p* values were calculated using Student’s two-tailed *t*-test with *p* < 0.05 considered statistically significant (Marked with *). Error bars represent means ± standard deviation (SD) (n = 3). (**c**) Peroxisome localization in FBFP-SKL strain. SKL is a peroxisome localization peptide. (**d**) Percent of cells with peroxisome in vacuole. *p* values were calculated using Student’s two-tailed *t*-test with *p* < 0.05 considered statistically significant (Marked with *), and ** represent extremely significant difference (*p* < 0.01). Error bars represent means ± standard deviation (SD) (n = 3).

**Figure 7 molecules-28-03041-f007:**
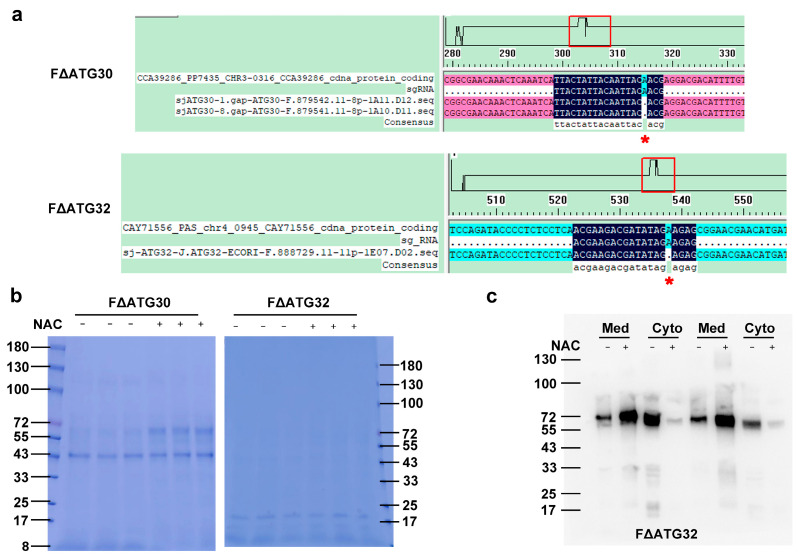
Effect of *ATG30* and *ATG32* gene disruption on NAC-promoted HSA-pFSHβ secretion. (**a**) Shift mutations of *ATG30* and *ATG32* in strains of FΔATG30 and FΔATG32 were identified by sequencing. Red * represent the deleted base in *ATG30* or *ATG32* gene of F strain. (**b**) SDS-PAGE analysis of recombinant HSA-pFSHβ protein in culture medium of FΔATG30 and FΔATG32 strains treated with 5 mM NAC for 72 h. +: represent culture medium supplemented with NAC, otherwise marked -. (**c**) Western blot analysis of intracellular HSA-pFSHβ fused proteins. Med represents HSA-pFSHβ protein in culture medium, cyto represents intracellular HSA-pFSHβ protein.

**Table 1 molecules-28-03041-t001:** Recombinant strains of *Pichia pastoris* in this study.

Strains Name	Description	Genotype	Source
F strain	WT	GS115 HSA-pFSHβ::prAOX(Zeocin^r^) his4	
FATG8	WT/pr*GAP*-yEGFP3-ATG8	F yEGFP3-ATG8::GAP his4	This study
FApe1	WT/pr*ACT*-Ape1-yEGFP3	F Ape1-yEGFP3::ACT his4	This study
FTom20	WT/pr*TOM20*-Tom20-yEGFP3	F Tom20-yEGFP3::TOM20 his4	This study
FBFP-SKL	WT/pr*GAP*-BFP-SKL	F BFP-SKL::GAP his4	This study
FΔATG30	atg30Δ	F atg30Δ::Hygromycin Br his4	This study
FΔATG32	atg32Δ	F atg32Δ::Hygromycin Br his4	This study
FTom20-ΔATG32	FTom20/ΔATG32	FTom20 atg32Δ::Hygromycin Br his4	This study
FApe1-ΔATG30	FApe1/ΔATG30	FApe1 atg30Δ::Hygromycin Br his4	This study

## Data Availability

The original data in the article can be obtained directly from the corresponding author.

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
