# Peer review of "Effect of N-acetyl-l-cysteine on Cell Phenotype and Autophagy in Pichia pastoris Expressing Human Serum Albumin and Porcine Follicle-Stimulating Hormone Fusion Protein"

_molecules, 2023, doi:10.3390/molecules28073041_

Round 1

Reviewer 1 Report

Comments to author:

This manuscript focused on the “Effect of N-acetyl-L-cysteine on cell phenotype and autophagy in Pichia pastoris expressing Human serum albumin and porcine Follicle-stimulating hormone fusion protein
”. As a general impression, this study is very interesting and well written. I have few minor comments:

1.     Line 82, what is YPD, need to elaborate when using first time in the manuscript.

2.     What is the difference between YPD and F strain, Why F strain was chosen? please mention in Result section 2.1.

3.     Figure 2: Needs better resolution, cannot read labels.

4.     Lines 140-142, need citation to show the mentioned genes are involved in pexophagy or mitophagy.

5.     Likewise, citation is needed for statement made in lines 144-146.

6.     It would be better to show Table 1. in the supplementary section.

7.     Figure 7a: Needs better resolution 

Author Response

Author’s response to the issues raised by reviewer 1

1. Line 82, what is YPD, need to elaborate when using first time in the manuscript.

Response: Thank you very much for your comment to improve our manuscript. We are so sorry for our negligence. As your suggestion, we have added the full spelling of YPD (line 83 of page 2) and have checked the full text to prevent similar errors in the revised manuscript marked with blue.

2. What is the difference between YPD and F strain, Why F strain was chosen? please mention in Result section 2.1.

Response: Thanks so much for your suggestion to improve our manuscript. We are so sorry for our negligence. As you suggestion, we need to elaborate on F strain which is a recombinant GS115 strain that expresses HSA-pFSHβ protein at a high level. YPD (Yeast extract Peptone Dextrose medium) is a medium for the cultivation of yeast. To avoid unnecessary ambiguity, we have defined the F strain on line 81 of page 2 in the revised manuscript.

3. Figure 2: Needs better resolution, cannot read labels.

Response: Thank you very much for your comment to improve our manuscript. As your suggest, we have enlarged Figure 2c, so that the gap between the cell membrane and the cell wall can be seen more clearly in the revised manuscript.

4. Lines 140-142, need citation to show the mentioned genes are involved in pexophagy or mitophagy.

5. Likewise, citation is needed for statement made in lines 144-146.

Response: Thanks so much for your comments to improve our manuscript. We apologized for our negligence. Based on your comments, we have cited the corresponding literature when we first mentioned the function of genes on lines 151, 153 and 156 and organized the literature in the revised manuscript.

6. It would be better to show Table 1. in the supplementary section.

Response: Thanks so much for your suggestion to improve our manuscript. Based on your comments, we have putted Table 1 in the revised supplementary material.

7. Figure 7a: Needs better resolution

Response: We apologize for the inconvenience caused by the small size of the picture, which makes it unclear. To make the image clearer, we have enlarged Figure 7a and provided an image with a higher resolution.

Reviewer 2 Report

Here, the authors address the mechanism of N-acetyl-L-cysteine (NAC)-induced secretion of human serum albumin and porcine follicle-stimulating hormone fusion protein (HSA-pFSHβ) in Pichia pastoris. NAC decreased the growth rate and altered the cell wall composition. Decreased expression of genes related to intracellular transport and autophagy was also verified. The levels of autophagy, mitophagy, and pexophagy were increased. These findings contribute insight into the mechanisms of protein secretion in Pichia pastoris, with a potential impact in increasing its recombinant protein purification yield.

Revision recommendations:

1. Please mention the full name of YPD in section 2.1.

2. Please briefly introduce the F strain in the results section. Please also mention the rationale for using the Δyps 1 strain in section 2.1. What protein is deleted in this strain?

3. Please mention whether the p-values are one-tailed or two-tailed.

4. Figure 1a, please change the order of CFW and CR columns, to be consistent with panels b and c.

5. In figure 2c, please re-check if both images are at the same magnification.

6. In figure 3, please indicate which bars refer to NAC or control.

7. In figure 4, panel "e" is labeled as panel "f". Please correct in the figure, legend, and main text.

8. In figure 7a, please provide an image with a higher resolution.

9. I suggest placing the materials and methods section after the conclusion section.

10. In section 4.5, please indicate the % polyacrylamide in SDS-PAGE gels.
